# Experimental total uncertainty of the derived GNSS-integrated water vapour using four co-located techniques in Finland

Ermanno Fionda<sup>1</sup>, Maria Cadeddu<sup>2</sup>, Vinia Mattioli<sup>3</sup>, Rosa Pacione<sup>4</sup>

5<sup>1</sup> Fondazione Ugo Bordoni (FUB), Roma (Italy)

<sup>2</sup> Argonne National Laboratory (ANL), Lemont, IL (USA)

<sup>3</sup> CETEMPS, L'Aquila (Italy)

<sup>4</sup> e-GEOS Spa, Matera (Italy)

Correspondence to: Ermanno Fionda (efionda@fub.it)

Abstract. In this work, we examine data from a Global Positioning System (GPS) ground-based receiver, two co-located ground-based microwave radiometers (MWRs), and radiosondes (RAOBs) to characterize the uncertainties associated with

- integrated water vapour (IWV) values estimated from the GPS in a sub-Arctic climate region. The experiment was carried out during the Biogenic Aerosols–Effects on Clouds and Climate research campaign conducted using the Atmospheric Radiation Measurement Program's second Mobile Facility (AMF2) in collaboration with the University of Helsinki. The GPS receiver was located about 20 km away from the AMF2 instruments (radiometers and RAOB). The GPS data were processed in Precise Point Positioning mode using the state-of-the-art scientific software GIPSY-OASIS II. Differences between the GPS-derived
- IWV and that derived from the other three instruments are analysed in terms of bias, standard deviation, and root-mean-square error (RMSE). The availability of three co-located, independently calibrated systems (two MWRs and one RAOB) allows us to isolate issues that may be specific to a single system and to isolate the effects of the distance between the GPS receiver and the remaining instruments. The representativeness error due to the 20-km distance between the GPS and the other systems is of the order of 0.6–1.5 kg/m<sup>2</sup> and in this study is the dominant effect when the IWV is higher than 20 kg/m<sup>2</sup>. The RMSE
- between the instruments shows that in the sub-Arctic region, when the IWV variability is less than 20 kg/m<sup>2</sup>, the GPS agrees with other instruments to within 0.5 kg/m<sup>2</sup>. When the variability of water vapour in the 20-km region is higher than 20 kg/m<sup>2</sup>, mostly in the summer months, the GPS agrees with other instruments within 1–2 kg/m<sup>2</sup>.

### **1** Introduction

Although water vapour accounts for only about 0.25% of the total mass of the atmosphere, it determines most of the Earth's 30 energy budget and large-scale circulation. This critical and fundamental tropospheric component affects cloud distributions, storm initiation, and heat transport from the surface to upper layers, and it is therefore relevant for the thermodynamic properties of the entire atmosphere. Therefore, knowledge of spatial and temporal distribution of water vapour is important for a variety of applications. Since about 1930, regular observations of moisture profiles have been possible,

thanks to the radiosonde (RAOB), and today a variety of instruments monitor water vapour at various temporal and spatial scales. Among these techniques, Global Positioning System (GPS) observations are becoming more important, with the fast growth of ground network receivers making GPS a very crucial and innovative system for water vapour monitoring. The estimate of integrated water vapour (IWV) from GPS is possible thanks to the linear relationship between the zenith wet delay

- (ZWD) and the IWV present in the volume of atmosphere traversed by the signal from space to the ground receiver. Several studies conducted in various climatic regions of the Earth have evidenced weaknesses and strengths of the GPS by comparing GPS measurements with measurements from RAOBs, multi-channel microwave radiometers (MWRs), infrared radiometers at the ground or aboard satellites, the Very Long Baseline Interferometry (VLBI) radio telescope, and numerical weather prediction models (Van Maldern et al. 2014; Choy et al., 2015; Memmo et al., 2005). From a statistical point of view, inter-
- comparison analyses have highlighted discrepancies in terms of systematic errors (bias) and random errors or root-mean-square error (RMSE). These discrepancies can be attributed to different factors related to climatic conditions, length of observations, characteristics of the instruments used such as instrumental errors, volumes sampled, sensitivity, sampling time, etc., as well as to the water vapour retrieval algorithms. Inter-comparing and quantifying reprocessed ground-based GPS tropospheric datasets against IWV and zenith total delay (ZTD) datasets from independent geodetic techniques (VLBI and Doppler
- Orbitography Radio Positioning Integrated by Satellite) and atmospheric in situ and remote sensing techniques (such as RAOBs, MWRs, sun photometers, and satellite water vapour products) are two of the objectives of the "Working Group 3: GNSS for climate monitoring" of the EU COST Action ES1206, "Advanced Global Navigation Satellite Systems tropospheric products for monitoring severe weather events and climate (GNSS4SWEC)," launched for the period of 2013–2017. A comprehensive review of such comparisons is available in the Final Report of the COST ES1206 (COST Action ES1206,
- 2018). However, because the observation technologies rapidly improve, it is useful to periodically repeat comparison campaigns between datasets, as mentioned in the World Climate Research Program Report 19/2015 (Schulz et al., 2015). In studies on the sub-Arctic climate, a recent comparison by Berezin et al. (2016) reported RMSEs ranging from 1.84 to 3.10 kg/m<sup>2</sup> and systematic differences/errors (bias) from 0.37 to 1.24 kg/m<sup>2</sup> during a campaign conducted near St. Petersburg (Russia). Ning et al. (2012) compared 10 years of wet delay estimates in Sweden and found differences of a few kg/m<sup>2</sup> among
- the various techniques. Buehler et al. (2012) compared six datasets in Sweden and, although they found overall good agreement between the measurements (with systematic differences below 1 kg/m<sup>2</sup>), they also highlighted the dependence of the results on the specific instrument. Another interesting finding was that the representativeness errors dominate the random error between the datasets and therefore they are not indicative of the true precision of the technique. Building on these two latter studies, we present here results from a comparison campaign conducted in Hyytiälä, in the South Region of Finland in 2014.
- Data were collected during the Biogenic Aerosols-Effects on Clouds and Climate (BAECC) research campaign, conducted using the Atmospheric Radiation Measurement (ARM) Program's second Mobile Facility (AMF2) in collaboration with the University of Helsinki. During the campaign, three collocated instruments (two MWRs and one RAOB) provided estimates of water vapour at the measurement site for almost one full year. These measurements are compared with data from the GPS receiver located in Orivesi, about 20 km from the ARM AMF2 site.

The purpose of the present study is to contribute to the understanding of the differences between observation systems as described by Buehler et al. (2012), by leveraging some of the unique features of the BAECC campaign: The first unique feature was the presence of two MWRs, one of which was a 3-channel radiometer that included the 89-GHz frequency, which enhances the sensitivity of the instrument to small amounts of water vapour. Having two co-located, independently calibrated

- radiometers that provide measurements at different frequencies can help identify uncertainties in the calibration, as well as eventual drifts. On the other hand, the presence of three systems (two MWRs and GPS) in addition to the RAOB can help isolate issues associated with the RAOB dry bias. Lastly, the spatial distance between the GPS and the remaining instruments can provide some insights into the representativeness errors and help to isolate them from the true random uncertainty of the systems.
- The methodology used in the study is the following: First, the water vapour estimate is derived from each of the instruments independently and the noise characteristics of the MWRs and GPS are examined at the highest temporal resolution to assess the expected instrumental noise. Second, the measurements are all averaged for 15 minutes around the RAOB launch time. A dataset of coincident measurements is therefore produced. As a third step, the GPS is taken as a reference and scatter plots of the corresponding measurements are shown, providing statistical information on the overall agreement between the
- measurements. To better understand the comparison, we analyse separately the bias and standard deviation (SD) of the differences and show the RMSE between the instruments. The results of these analysis are presented and results are summarized.

#### 2 Dataset description

In this section, we provide a brief overview of the measurement sites and datasets used in this work.

#### 20 2.1 The AMF2 observation site

The ARM AMF2 deployment was installed in Hyytiälä ( $61^{\circ}51^{\circ}N$ ,  $24^{\circ}17^{\circ}E$ ) between January and September 2014, during the BAECC campaign (http://www.arm.gov/sites/amf/tmp). Two MWRs were deployed as part of the ARM Program's Mobile Facility (Figure 1). One radiometer (MWR2C) operates at two frequency channels (*f*) at 23.8 and 31.4 GHz; the second (MWR3C) has three channels at 23.8, 30.0, and 89.0 GHz. Although the two systems are fairly similar, there are some

- differences that are important for the present comparison, First, the two systems have different fields of view (FOVs); the MWR2C has a 6-degree FOV (HPBW) for both channels and the MWR3C has a 3-degree FOV for all channels. The time sampling is also different. The MWR2C samples at irregular times, alternating faster zenith sampling (every 20 s) during cloudy conditions with sampling every ~50 s during clear sky conditions. The MWR3C collects zenith observations at 10-s intervals and performs tip scans at regular intervals of about 10 minutes. The most important difference, however, is the
- presence of the 89-GHz channel, which increases the MWR3C sensitivity to water vapour, making it a good instrument for locations at high latitudes. For a description of the radiometers and their calibration, see Cadeddu et al. (2013). Although the

random uncertainty of the radiometric brightness temperature  $(B_T)$  is expected to be very low, of the order of 0.1 degree K, there are other factors that can affect the accuracy of long-term measurements. Drifts in the noise diode as well as thermal effects on the radiometric components that are not thermally stabilized can introduce seasonal and diurnal biases. The MWR2C and MWR3C differ in their thermal stabilization and in the way they handle the correction of residual temperature effects on

- the calibration. Additional biases can be introduced in the calculation of the opacity by the estimation of the mean radiating temperatures used in the tip calibration. The ARM radiometers are calibrated with a high degree of confidence and are expected to have a measurement RMSE of around 0.3 K in the K-band and 0.5 K in the W-band (89 GHz). The uncertainty in retrieved IWV from the 2-channel radiometer is expected to be 0.5 kg/m<sup>2</sup>; however, with the addition of the higher frequency, it is possible to reduce the IWV uncertainty to about 0.4 kg/m<sup>2</sup>.
- Although the ARM program provides routine retrievals of IWV and liquid water path, a specific retrieval algorithm for water vapour was developed in this work. For this purpose, several regression algorithms were tested by using opacities  $\tau(f)$  and B<sub>T</sub>(*f*) at the sampled frequencies as predictors. The algorithms use various polynomial combinations of the predictors and different computations of the mean radiating temperature (Wu, 1979; Westwater et al., 1980; Basili et al., 1998) are used to calculate the predictor  $\tau(f)$ . Retrieval coefficients were derived using a long-term RAOB database collected in Jyväskylä and
- the performances of the selected algorithms were compared. The Jyväskylä database encompasses 12 years (2003–2014) of RAOB profiles launched about 90 km away from the ARM AMF2 site. The profiles were adjusted for the site altitude with respect to the ARM AMF2 site altitude. All simulations were carried out by applying the Radiative Transfer Equation, together with a cloud model, to the RAOB database (Ulaby et al., 1981 Mattioli et al., 2009). The retrieval algorithm based on a linear combination of B<sub>T</sub>(*f*) was found to give the best agreement with the "true" IWV in the testing subset of profiles, with a relatively
- small standard SD of 0.72 kg/m<sup>2</sup> and a correlation coefficient of 0.997. This algorithm (Eq. 1) was therefore chosen to retrieve IWV from the AMF2 site measurements used in the next section:

$$IWV = a_0 + \sum_{i=1}^n b_i B_T(f_i) \quad , \tag{1}$$

where n is the number of radiometric frequency channels (n=3 for the MWR3C and n=2 for the MWR2C), and  $a_0$ ,  $b_j$  represent the computed linear inversion coefficients. The linear relationship of IWV to  $B_T(f)$  holds for the very low  $B_T(f)$  temperatures corresponding to the low IWV values found in the Arctic. As IWV increases, linearity holds with atmospheric opacities (Westwater et al., 1980).

During the BAECC research campaign, atmospheric profiles were collected by RAOBs, model RS92, launched four times a day (at 00, 05, 11, and 17 UTC) near the MWR stations. The performance of the RS92 humidity sensor has been the subject of several investigations. In particular, Wang and Zhang (2008) have observed that RS92 profiles could be affected by solar radiation dry bias (dependent on pressure, season, and time of day), which requires a relative humidity correction. RAOB dry bias has been extensively documented for example, by Miloshevich et al. (2009), who estimated a precision of about 5% for the RAOB-derived IWV, and by Cady-Pereira et al. (2008), who proposed a correction dependent on the solar zenith angle.

25

More recently, Wang et al. (2013) proposed corrections of the order of 2% to 4% in the lower mid-troposphere and 6% to 8% in the upper troposphere.

# 2.2 GPS data

- The GPS ground-receiver is located at the Orivesi station, about 20 km away from the AMF2 site. The GPS station is operated 5 by the Finnish Geodetic Institute and the National Land Survey of Finland. GPS data were obtained as RINEX files and processed in Precise Point Positioning mode using GIPSY-OASIS II and state-of-the-art processing options to derive ZTD with a 5-minute sampling (Pacione et al., 2014; Pacione et al., 2017). The ZTD known as the "tropospheric total delay" by the neutral atmosphere can be expressed as the sum of two components: zenith hydrostatic delay (ZHD) due to hydrostatic gases (mainly oxygen) and ZWD due to water vapour (Elgered, 1993). In this work, ZTD is used to estimate ZWD, which is, in turn,
- related to the atmospheric IWV according to Eq. 2:

$$ZWD = ZTD - ZHD$$
(2)

The ZHD used in Eq. 2 can be accurately modelled from knowledge of the total surface air pressure, latitude, and altitude of

- the Global Navigation Satellite Systems (GNSS) ground-receiver station above the geoid (Saastamoinen, 1972, Davis et al., 1985). The error introduced by assuming hydrostatic equilibrium is typically of the order of 0.01%, which corresponds to a sub-millimetre accuracy in zenith delay. Nevertheless, under extreme conditions, ZHD errors can amount to several mm (Elgered, 1993). The uncertainty in the calculation of ZHD is mainly due to uncertainty in surface pressure, refractive constant, and dry gas constant (Ning et al., 2016). An error in surface pressure of 1 hPa will lead to a 2.3 mm error in ZHD and about
- $0.35 \text{ kg/m}^2$  in IWV (Elgered et. al., 1991; Davis et al, 1895). Because a pressure sensor is not available at the GPS site, ZHD was computed using the air pressure provided by the RAOBs launched at the AMF2 site, adjusted for the different site altitudes. Assuming that the wet path delay is entirely due to the amount of water vapour traversed by the signal, IWV can be computed from ZWD via a dimensionless conversion factor  $\pi$  (Bevis et al., 1992; Bevis et al., 1994; Rocken et al., 1993; Rocken et al., 1995; Duan et al., 1996; Tregoning et al., 1998; Davis et al., 1985), as shown in Eq. 3:

$$IWV = \pi * ZWD \tag{3}$$

As can be seen from Eqs. 2 and 3, uncertainties in GPS-derived IWV derive from uncertainties in ZTD processing as well as uncertainties in  $\pi$  and ZHD. Ning et al. (2012, 2016) assessed that, because ZHD can be estimated fairly accurately, ZTD

processing contributes about 75% to the total IWV uncertainty. The dimensionless factor π is a function of various physical constants and of the weighted mean atmospheric temperature (Tm) (Askne and Nordius, 1987; Davis et al., 1985; Duan et al., 1996). Tm is commonly computed from surface air temperature using a linear regression (Bevis et al., 1992) with an RMSE

of 2-5 K. Tm is generally site-dependent and varies seasonally and diurnally (Mattioli et al., 2005; Wang et al., 2005). In this study, the time-varying  $\pi$  is statistically computed from a linear regression between IWV and ZWD. Specifically, the same large RAOB database collected in Jyväskylä that was used to derive the radiometric coefficients (as described in section 2.1) was also used to compute the ZWD from the profiles. This method seems to fit the sub-Arctic climatic conditions well,

introducing low relative errors in IWV (Basili et al, 2001). As expected, the computed  $\pi$  displays a seasonal variability, with a maximum value in July (0.158) and a minimum value in February (0.148). The contribution of  $\pi$  to the IWV uncertainty is evaluated as

$$\sigma_{IWV} = \sigma_{\pi} * \overline{ZWD} \quad , \tag{4}$$

where  $\sigma_{\pi}$  is the monthly calculated SD of the conversion factor and  $\overline{ZWD}$  is the monthly averaged ZWD. To express this uncertainty as a percent,  $\sigma_{IWV}$  can be divided by the monthly mean IWV. The resulting IWV uncertainty varies from 1.15% (July) to 1.83% (February) and is lower than the 2.8% error obtained when using a single annual value, as shown in Table 1. The use of monthly values leads to a reduction in the IWV uncertainty of between 34% and 58%, depending on the month. All

15 the uncertainties mentioned above, with additional factors due to multipath effects (Pierdicca et al., 2014), contribute to the total uncertainty of the GPS-derived IWV.

# 3 Comparison between the four instruments

In this section, we compare the IWV derived from the four deployed systems: GPS, MWR3C, MWR2C, and RAOB. There are several ways in which differences can be expressed when two or more systems are compared. Keeping in mind that each system has its own intrinsic uncertainties, discussed in section 2, differences between measurements will also depend on different temporal sampling and spatial representativeness. In the present study, for example, the GPS is located about 20 km away from the site, the RAOBs are launched at the site but quickly drift away, and the two radiometers sample in zenith view with different FOVs. A general level of agreement between measurements can be achieved by regression analysis, i.e., by examining the correlation between the measurements throughout the campaign. A regression analysis can help in the 25 identification of biases or dependencies of the differences on factors such as water vapour amount or temperature. Another

way uncertainties can be characterized is by examining the differences D between the measurements defined as the mean value of the differences between the GPS and the other instrumentals (x) on the N experimental samples,

$$D_{GPS,x} = \frac{1}{N} \sum_{1}^{N} IWV_{GPS} - IWV_x \quad , \tag{5}$$

30

the SD of the differences,

(6)

$$SD_{GPS,x} = \sqrt{\frac{1}{N-1} \sum_{1}^{N} (IWV_{GPS} - IWV_{x} - D_{GPS,x})^{2}}$$
,

and their mean square differences (RMSD):

$$RMSD_{GPS,x} = \sqrt{\frac{1}{N} \sum_{1}^{N} (IWV_{GPS} - IWV_{x})^{2}}$$
 (7)

It can be shown that Eq. 7 is related to Eq. 5 and 6 by  $RMSD^2 = D^2 + SD^2$  and is a total measure of the systematic and random variability of the difference between the two measurements. A 3.3-hour segment of concurrent IWV data collected during the night of March 26, 2014, is shown in Figure 2. The mean value of each segment was subtracted to calculate the short-term

random variability of each instrument at the highest temporal resolution. The SD of the MWR3C (blue) and the MWR2C (black) retrievals over the 3.3 hours is about 0.2 kg/m<sup>2</sup>. The temporal resolution of the GPS is much coarser; only 40 samples are collected in 3.3 hours, with a SD of 0.15 kg/m<sup>2</sup>. In the following two sections, the agreement among the four instruments is examined with a focus on the GPS-derived IWV.

### 15 3.1 Regression analysis

The IWV derived from the MWR3C, MWR2C, and GPS was averaged over 15 minutes starting at the time of the RAOB launches. After eliminating radiometric data that were affected by poor calibration at the beginning of the campaign and after screening out data contaminated by precipitation, a total of 424 samples were kept for further analysis. When comparing the

- GPS data with the other instruments' data, the different altitude of the GPS site was accounted for. The entire time series of the RAOB-matched IWV values observed during the campaign is shown in Figure 3. From Figure 3, it can be seen that the IWV at the site is fairly low (less than 10 kg/m<sup>2</sup>) during the winter months, as expected, and it gradually increases through the spring and summer months to reach a maximum value of about 35 kg/m<sup>2</sup> in July. Specifically, the 2014 IWV from the RAOB showed a lognormal distribution with an average value of about 12.8 kg/m<sup>2</sup>, a median of 10.7 kg/m<sup>2</sup>, and a mode of 8.2 kg/m<sup>2</sup>.
- From an inter-comparison point of view, the four independent systems agreed fairly well throughout the 2014 BAECC research campaign. Figure 4 shows scatter plots of the GPS (Y-axis) vs the MWRs and RAOB (X-axis). The vertical error bars represent the 15-minute variability of the GPS data.

From Figure 4, it is evident that a generally good agreement exists between the GPS and the other 3 instruments, with very low differences and SD of the order of 1 kg/m<sup>2</sup>. MWR3C has the smallest differences (-0.004 kg/m<sup>2</sup>) and the RAOB and

30 MWR2C have differences of about 0.2 kg/m<sup>2</sup>. The remaining statistical parameters are fairly similar for all the instruments, with perhaps the exception of the slope and intercept of MWR2C. The 2-channel radiometer seems to slightly underestimate when the IWV < 15 kg/m<sup>2</sup> and overestimate when the IWV > 15 kg/m<sup>2</sup>, leading to a higher intercept and lower slope than in

the other two systems. As discussed, the enhanced sensitivity of the 89-GHz channel to small IWV amounts makes MWR3C more suitable for this climatic region, and the instrument provides the overall best agreement with the GPS data. To better understand the nature of the differences between the systems, the next section analyses the differences and SD separately.

# 5 3.2 Differences and standard deviation

Differences and SDs between GPS and the other 3 co-located instruments are shown in Figure 5 as black circles and black diamonds. The differences are dominated by the SD, which is of the order of  $1 \text{ kg/m}^2$  for all instruments. In particular, the GPS and MWR3C are unbiased and their differences are entirely due to random variability, namely, the instrument's random noise,

- IWV variability in the instrument's FOV, and representativeness errors due to the spatial distance (20 km) between the sites. In fact, if we examine the differences and SD between the MWR3C and the 2 co-located systems, MWR2C and RAOB, shown in Figure 5 as red circles and diamonds, respectively, we see that the SD is about 0.5 kg/m<sup>2</sup>. This suggests that spatial distance between the GPS and the other instruments contributes to about half of the total random variability. This aspect will be discussed in more detail later.
- To investigate the presence of diurnal effects on the differences, we display in the top panel of Figure 6 the differences (mean differences) between the GPS and other systems at the four times of RAOB launches. The top panel of Figure 6 shows that the difference between GPS and MWR3C is very low throughout the day and does not display a specific diurnal trend. The MWR2C and RAOB, on the other hand, have diurnal behaviors. In the case of MWR2C, the difference is higher at night and decreases during the day, indicating a possible residual dependence of the calibration on the temperature; in the case of the
- RAOB, the difference is largest at 11 UTC, indicating that the RAOB dry bias can be affecting the measurements. This possibility is further investigated by calculating the differenceses of all instruments with respect to the RAOB. These are shown in the bottom panel of Figure 6, where the differences between all systems and the RAOBs are shown at the four times of RAOB launches. The increased differences at 11 UTC indicate that the RAOB dry bias is affecting the comparison. Therefore, in the remainder of this study, the 11 UTC data are eliminated to eliminate the effect of the dry bias on the results. Because
- the MWR2C seems to display a diurnal variability as well, we limit our consideration to only the GPS, MWR3C, and RAOB in the rest of the study.

The IWV differences as a function of IWV are shown in Figure 7 for the GPS, MWR3C, and RAOB. The top panel shows the IWV differences between GPS and RAOB and the middle panel those between GPS and MWR3C as a function of IWV. Averages and SDs for IWV bins of 5 kg/m<sup>2</sup> are over-plotted as squares with error bars. In the two top panels, the SD increases

with increasing IWV from 0.3 kg/m<sup>2</sup> to about 2 kg/m<sup>2</sup>. Such an increasing trend is not observed when the co-located instruments are examined (bottom panel of Figure 7). In this case, the SD of the differences between MWR3C and RAOB is around 0.3–0.5 kg/m<sup>2</sup> in all IWV bins. The increased random variability between GPS and the other instruments for higher IWV amounts is likely due to the spatial variability of the water vapour in the 20-km area that separates the sites. This variability accounts for more than half of the total random uncertainty. To further investigate this point and to assess whether the increase in the

differences between GPS and other instruments may be due the instruments own variability we show in the left panel of Fig. 8 the RMSD between the instruments and, in the right panel of Fig. 8, the standard deviation of each instrument in each IWV bin. As mentioned earlier the RMSD (eq. 7) is a measure of total (systematic and random) differences between two systems while the standard deviation of each instrument about the mean is a measure of the individual instrument variability. The left

- panel of Figure 8 shows the IWV RMSD between GPS and MWR3C (black), GPS and RAOB (red), and MWR3C and RAOB (green). For consistency all differences were binned using the MWR3C IWV. On the basis of the previous discussion, it can be seen that when the IWV is low, the total differences between GPS and RAOB and MWR3C are around 0.5–0.6 kg/m<sup>2</sup>. This is in good agreement with the theoretical estimate of Ning et al. (2016), who, using error propagation, estimated a total uncertainty of ~0.6 kg/m<sup>2</sup>. When the IWV increases the differences between GPS and other instruments progressively increase
- to reach about 2 kg/m<sup>2</sup>. This is not observed in the differences between co-located instruments (MWR3C and RAOB). The right panel of Fig. 8 shows the mean (x-axis) and the standard deviation (y-axis) of the instruments in each IWV bin. From the right panel of Fig. 8 it can be seen that the variability of the single instruments about the mean, does not display a marked dependence on the IWV. In addition, all instruments display similar random variability about the mean. It is arguable that the increase in the differences between the non-collocated instruments with IWV is then due to the spatial distance between the
- systems. By looking at Fig. 2 it can be seen that IWV above 20 g/m<sup>2</sup> is a frequent occurrence in summer when higher spatial variability of IWV can be expected. If GPS measurements are used to represent large geographical areas, it is important to include this larger random uncertainty in the analysis due to the spatial variability of water vapor. Our study suggests that in this climatic region, the spatial representativeness error increases with the water vapour amount and reaches about 1–2 kg/m<sup>2</sup> when the water vapour exceeds 20 kg/m<sup>2</sup>.

#### 20 4 Conclusions

In this work, nine months of IWV data derived from four instruments located in South Finland were compared. Three instruments (two MWRs and one RAOB) were part of the ARM AMF2 deployment and collected data in Hyytiälä between January and September 2014, during the BAECC campaign. A GPS receiver was located at the Orivesi station, about 20 km away from the AMF2 site. The IWV was derived from each instrument and all measurements were averaged for 15 minutes at the times of the RAOB launches (00, 05, 11, 17 UTC).

A few important aspects of this study are highlighted: First, the presence of two independently calibrated MWRs gives confidence in the stability of the radiometric measurements, and the addition of the 89-GHz channel increases the sensitivity of the measurements to small amounts of IWV. Second, the presence of multiple instruments enables us to isolate features associated with one particular system, such as the dry bias in the RAOB measurements and a weak residual temperature

dependence in the MWR2C. Finally, the 20-km distance between the GPS and the other three instruments helps identify the contribution to the random variability due to the spatial inhomogeneity of the water vapour.