# Peer review of "Experimental total uncertainty of the derived GNSS-integrated water vapour using four co-located techniques in Finland"

_Atmospheric Measurement Techniques, 2018_

## Referee Comment (RC1) · Anonymous Referee #1 · 18 Aug 2018

General Comments

The manuscript presents a multisensor comparison of the Integrated Water Vapour (IWV) in the atmosphere using GPS, radiosondes, and two ground-based microwave radiometers. The focus is on the GPS, but also calibration of the microwave radiometers and the so called dry bias effect for the radiosondes are mentioned. The work does not (to my knowledge) present any new knowledge in terms of significantly different conclusions compared to previously published comparisons using these types of sensors. I characterise the manuscript as being the basis for a "confirmation paper". The data are unique and - as the authors also state in the manuscript - "it is useful

to periodically repeat comparison campaigns between datasets". Having said that I think that the manuscript is well structured and focused on the important issues. My only concern in terms of length of the text is the conclusions. There is no need to repeat many of the results, especially since it is a short paper. My suggestion is to keep the text on page 9 and delete the text on page 10 (some rewring on page 9 will be necessary because of this). You may want to modify the text on page 9 in order to specifically mention that the real differences between the two sites dominate the observed differences (as least during the wetter part of the year), depending how you handle the suggestions below.

I think there is a potential for improvement of the manuscript and some suggestions are given below. Specifically I think the true variability in the IWV between the GPS site and the site of the other sensors could be taken further, although I cannot predict how interesting the result(s) will be.

Specific comments

The abstract is unnecessarily long and includes introductory information. I think you can ignore to mention the name of the experiment as well as how the GPS data were processed. Such information is already, as it should be, in Section 2.

The 20 km distance between the GPS and the other instruments is in several aspects a disadvantage but it does allow to try to separate the true RMS difference between the sites and the instrumental errors. Together these two effects cause the observed differences and it would be more clear if you would refer to the total RMS differences between GPS and the other sensors rather than RMS Error (RMSE). Part of the observed differences between GPS and the other sensors are signals, not errors. Following the same thought I recommend not to use the term representativeness error but rather true differences, or something similar.

You state that the GPS and MWR data were averaged for 15 minutes around the RAOB launch time. It would make more sense to make the average for a period starting at

the lunch time because that is the time when the RAOB sampling starts and it needs many minutes to rise through the layers with the main part of the water vapour.

I miss information about bandwidths, system noise temperatures, and integration times for the different radiometer channels? Since the radiometers are main instruments in the comparisons I think such information shall be in the paper rather than just a reference to an older paper.

I am confused by the description of the retrieval uncertainty for the MWR. First you say that it is expected to be 0.5 kg/mˆ2 and with the addition of the high frequency channel it can be reduced to 0.4 kg/mˆ2 (page 5, lines 8-9). Thereafter, (same page line 20) you say that the best retrieval algorithm resulted in an SD of 0.72 kg/m2. I must have missed a crucial point, please explain.

You mention the dry bias correction when describing the RAOB data, but it is unclear if you applied any correction, and if so which one?

You present a short data segment from March 26, 2014, in Figure 2. Why did you select this specific period? Is it a typical period, or perhaps a period that is very stable (with low variability)?

You say that the different height of the GPS site was accounted for (page 7), but you do not explain how?

Technical Corrections

page 1, line 13: In this work, we examine –> We examine

page 1, line 26 (and many other places): there shall be a space between the value and the unit (also the % unit) according to SI rules.

page 3: you may want to mention that the 89 GHz frequency is not only more sensitive to water vapour, but also to liquid water (clouds) which is a potential problem. This text could perhaps make more sense in Subsection 2.1 where the MWR is described.

page 3, end of introduction: it is common practise to refer to the section numbers when the structure of the papers is presented. It is helpful for the reader.

page 3: A better title for Subsection 2.1 could be "Microwave radiometers and radiosondes" ? It will match the title of Subsection 2.2.

page 4, line 30: the lunch times seem strange? I expect 6 h between the launches?

page 5, line 17: in zenith delay –> in the zenith delay

page 7, line 29: low differences –> low mean differences ? smallest differences –> smallest mean differences ?

page 8, line 7: Differences –> Mean differences

page 8, line 10: representativeness errors due to –> the true differences caused by

page 9, line 22: one RAOB –> RAOBs ?

Figure 1: I suggest to remove the text from the picture frame and describe which instruments (from left to right) that are seen in the figure caption. You may also want to mention the third instrument, the one to the right?

Figure 3: It is very difficult to see more that one of the time series. An alternative would be to show only one time series for the IWV and present the others as as differences from this one in individual subgraphs below.

Figure 4: I would prefer to have the important quantitative information either in the figure caption or in a separate table, if you regard it as important.

Figure 7: Some of the red squares are hard to see. Plot them with larger symbols and perhaps a more light red colour. And since they are few, plot them on top of the black circles.

Figure 8: Mean and standard deviation of the measurement of each instrument –> RMS differences for each instrument pair vs the mean IWV.

— End of Comments

---

## Referee Comment (RC2) · Anonymous Referee #2 · 20 Sep 2018

Review of manuscript amt-2018-161 "Experimental total uncertainty of the derived GNSS-integrated water vapour using four co-located techniques in Finland" by Ermanno Fionda, Maria Cadeddu, Vinia Mattioli, and Rosa Pacione.

General comments

This study compares IWV estimates measured with 4 different instruments in Finland over a period of 9 months. Three of the instruments (2 microwave radiometers and a radiosonde system) are collocated and the fourth one (GPS) is distant by 20 km. The paper presents a statistical analysis of IWV differences and discusses the results in terms of inter-system biases and random errors including "representativeness errors". The study reports a dry bias in the radiosonde data during day, a bias source in the 2-channel radiometer data, and a tendency in the GPS IWV differences to increase when IWV increases. The latter effect is explained as "representativeness errors" due to the 20-km distance between the GPS receiver and the other systems. Emphasis is made on this result but two other factors are expected to contribute as well to the observed results: differences in the sensitivity and volume of sensed atmosphere between GPS and the other techniques and instrumental errors specific to each technique. Since these factors are not quantified, it seems difficult to assess the third one. A rigorous way of evaluating the impact of the distance between remote techniques requires using identical instruments at both ends. The discussion and conclusions of this study are impaired by this deficiency. To overcome this major weakness, the authors could consider the following options:

(a) The differences in IWV due to a 20-km distance can be evaluated from two identical instruments (e.g. a pair of GPS receivers) separated by approximately this distance in the same climatic region;

(b) The differences in IWV due to the different sensitivity and volume of sensed atmosphere between GPS and other techniques can be evaluated from co-located measurements (e.g. from past campaigns).

Specific comments

Give a clear definition of the "representativeness errors" in the introduction (Page 2 Line 27). Strictly speaking, it should include the differences in the sensitivity and volume of sensed atmosphere between GPS and other techniques and the instrumental errors specific to each technique (see e.g. Buehler, AMT, 2012).

The MWR retrieval algorithm used radiosonde data from a station located 90-km apart. Could there be a "representativeness error" impacting the results? Did you compare the operational radiosonde observations to those from the BAECC campaign used in this study?

Can the linearity limitations observed with the 2-channel radiometer be explained/corrected? A temperature dependence is mentioned in the conclusions section though it is not discussed earlier. Please discuss this point in the text.

Give more details about the BAECC radiosonde data. What is the vertical resolution of the profiles? How are they quality-controlled? Is any data processing/filtering or bias correction applied?

Since the radiosonde measurements were made with a RS92 sonde type, did you consider applying the GRUAN bias corrections?

How are the IWV data adjusted for the differences in altitude? (P5 L21)

Why is the GPS conversion factor (pi) used as a monthly mean value and not from a linear regression with 2-m temperature following the method of Bevis et al. 1992? The uncertainty in the GPS IWV estimates could probably be reduced applying the Bevis method or time-varying Tm estimates from a NWP model.

It seems to me that your GPS IWV data contain outliers. Figure 4 and Figure 7 show large deviations in the IWV differences which involve the GPS data. The outlying values should be removed from the analysis.

The central description and discussion of results in section 3 should be supported by a statistical analysis including specifically the contributions from atmospheric variability (temporal/spatial) and the different sources of differences (representativeness, etc.) between measurements (see e.g. Ning et al., AMT, Vol. 9, p79-92, 2016).

According to your interpretation of the results, you assume (implicitly) that the differences are random and independent (e.g. between GPS and MWR3C). Can you bring evidence of this?

It is not clear how you estimate the representativeness errors contribution (P8 L12-14; P9 L17-19) Given the above assumptions, the standard deviations should be added/subtracted quadratically.

Figure 8 left panel: it might be more interesting to plot the SD rather than the RMSD to support the above discussion.

In the conclusion section it is stated that the comparison of the two radiometers gives confidence in the "stability" of these measurements but this point is not discussed earlier in the paper. Can you elaborate a bit more on this?

Technical comments

P1 L25 and 26: you refer incorrectly to IWV "variability" above or below 20 kg/m2 in place of simply "IWV". Please correct.

P6 The quantity defined by equation (5) should be referred to as the "mean difference"

P7 L23-24: the comment about the log-normal PDF is not used later, this sentence can be removed.

P7 L25: the comment on the agreement between measurements is elusive. Please remove it.

P8 L9: the GPS and MWR3C are "unbiased" maybe you want to write "have a mean different close to zero". To prove that an instrument is unbiased requires to compare its observations to a calibrated reference standard. However, you may argue here that the instruments provide independent observations which may have specific biases, however it is unlikely that their biases would perfectly cancel out, so it is much more likely here that they are both unbiased.

P8 L15: "higher variability in summer": this is not obvious from Figure 9 right panel.

P8 L16-19: you suggest to include a "representativeness" error of 1-2 kg/m2 to account for the spatial variability of GPS measurements. How would you do that? Shouldn't this error depend on the distance to the GPS measurements?

P8 L20-23: the comment of the bottom panel of Fig. 6 doesn't add any further information about the radiosonde bias at 11 UTC. Remove this comment as well as the bottom panel of Fig. 6 which shows merely a combination of the lines plotted in the upper panel.

P9 L21-31: these two paragraphs are not useful in the conclusion section.

Figure 3: the series are superposed and cannot be distinguished. Maybe show only one series to illustrate the IWV variations over the study period.

Figure 4: it is common practice to draw the common reference data set (here GPS) on the x-axis. Please change the 3 plots.

It may be useful to add a scatter plots of RAOB vs. MWR3C and MWR2C vs. MWR3C.

---

## Author Comment (AC1) · 5 Nov 2018

**Response to interactive comment of Referee #1 on "Experimental total uncertainty of the derived GNSS-integrated water vapour using four co-located techniques in Finland" by E. Fionda et al.**

**Anonymous Referee #1**

**In black => referee observations**
**In red => our response**

General Comments
The manuscript presents a multisensor comparison of the Integrated Water Vapour (IWV) in the atmosphere using GPS, radiosondes, and two ground-based microwave radiometers. The focus is on the GPS, but also calibration of the microwave radiometers and the so called dry bias effect for the radiosondes are mentioned. The work does not (to my knowledge) present any new knowledge in terms of significantly different conclusions compared to previously published comparisons using these types of sensors. I characterise the manuscript as being the basis for a "confirmation paper". The data are unique and - as the authors also state in the manuscript - "it is useful to periodically repeat comparison campaigns between datasets". Having said that I think that the manuscript is well structured and focused on the important issues. My only concern in terms of length of the text is the conclusions.

We greatly appreciate the referee's very constructive and valuable comments. Below we have tried to address all the reviewer's comments (General, Specific Comments and Technical Corrections) and we modified the manuscript accordingly. All comments and technical corrections are included in the current submitted version of the manuscript.

There is no need to repeat many of the results, especially since it is a short paper. My suggestion is to keep the text on page 9 and delete the text on page 10 (some rewriting on page 9 will be necessary because of this). *You may want to modify the text on page 9* in order to specifically mention that *the real differences between the two sites dominate the observed differences* (as least during the wetter part of the year), depending how you handle the suggestions below.

Following the suggestion and agreeing with the comment, we have rewritten part of the text on page 9 and deleted some on page 10. Also, following suggestions of Reviewer #2 we have added some discussion of the instrument's FOVs (fields of view) and added 2 Tables (Tab 2 and 3).

I think there is a potential for improvement of the manuscript and some suggestions are given below. Specifically I think the true variability in the IWV between the GPS site and the site of the other sensors could be taken further, although I cannot predict how interesting the result(s) will be.

We thank the reviewer for the suggestion We added Fig. 3 And Table 2 and added discussion in section 3.1 to discuss specifically this point.

**Specific comment:**
The abstract is unnecessarily long and includes introductory information. I think you can ignore to mention the name of the experiment as well as how the GPS data were processed. Such information is already, as it should be, in Section 2.

We agree with the reviewer and have shortened the abstract and eliminated the information that is in section 2.

The 20 km distance between the GPS and the other instruments is in several aspects a disadvantage but it does allow to try to separate the true RMS difference between the sites and the instrumental errors. Together these two effects cause the observed differences and it would be more clear if you would refer to the total RMS differences between GPS and the other sensors rather than RMS Error (RMSE).

Following both reviewers' comments we have eliminated the RMSD and kept the mean differences and standard deviation of differences. The results are now referred to as "differences due to the field of view" and "differences due to the distance".

Part of the observed differences between GPS and the other sensors are signals, not errors. I recommend not to use the term representativeness error but rather true differences, or something similar.

We followed the reviewers' suggestion and eliminated the term representativeness error and now we refer to the differences due to the FOV and to the distance in addition to the true random uncertainty of the instruments.

You state that the GPS and MWR data were averaged for 15 minutes around the RAOB launch time. It would make more sense to make the average for a period starting at the lunch time because that is the time when the RAOB sampling starts and it needs many minutes to rise through the layers with the main part of the water vapour.

We have been imprecise in writing the sentence on pp. 3 line 12 of the manuscript. The average values were taken over a time interval of 15 minutes from the launch time of radiosonde for the reasons well expressed by the referee.
The new sentence is: "Second, the measurements are all averaged over 15 minutes starting at the launch time of RAOB".

I miss information about bandwidths, system noise temperatures, and integration times for the different radiometer channels? Since the radiometers are main instruments in the comparisons I think such information shall be in the paper rather than just a reference to an older paper.

We have added the information at the beginning of section 2.1. Both radiometers have bandwidth of 300 MHz for the 23.8 and 30 GHz and 1900 MHz for the 90 GHz. The system noise temperature is < 500 K for the K-band channels and < 1100 K for the W-band channel and the integration time is 1 s for all channels.

I am confused by the description of the retrieval uncertainty for the MWR. First you say that it is expected to be 0.5 kg/m^2 and with the addition of the high frequency channel it can be reduced to 0.4 kg/m^2 (page 4, lines 8-9). Thereafter, (same page line 20) you say that the best retrieval algorithm resulted in an SD of 0.72 kg/m^2. I must have missed a crucial point, please explain.

The quoted uncertainty in the retrieval (0.5 kg/m$^2$ for a two-channel and 0.4 kg/m$^2$ for a 3-channel retrieval) is an estimated theoretical uncertainty derived by accounting for the calibration uncertainty combined with the sensitivity of each channel to water vapour (intended as the slope of TB vs PWV, see for example Racette et al., 2005, now added in the reference list). Assuming a calibration uncertainty of 0.3 K for the K-band channels (23.8 and 31 GHz) and 0.5 K for the W-band channel (89 GHz) the uncertainty in the water vapour can be derived as DPWV=DTb/slope. Because we are in the linear regime of the sensitivity we can follow Racette et. al. for the estimated slopes (1.25, 0.34,

and 1.8 K/mm) and estimate the uncertainty of PWV derived from each single channel as 0.24, 0.88, and 0.28 kg/m$^2$. [Racette et al., Measurement of Low Amounts of Precipitable Water Vapor Using Ground-Based Millimeter wave Radiometry, J. Atmos. Oceanic. Techn., 22, 5, 317-337, 2005]. We added text in section 2.1 to explain this.

The SD=0.72 kg/m$^2$ refers to the minimum IWV SD produced by the "best" selected retrieval algorithm derived from a statistical dataset of radiosonde from the Jyväskylä station.
The retrieval algorithm consisting of a set of retrieval coefficients of a polynomial equation able to predict IWV from MWR observations was derived from a simulated training dataset, that includes clear-sky and cloudy conditions, and its performance is evaluated by applying it to a simulated independent test dataset.

The total uncertainty of the retrieval will depend on:
-intrinsic variability of the used dataset;
-how well the dataset represents the climate of the region under analysis;
-RTE used to derive simulation also including a cloud model;
-predictors;
-polynomial expressions.

You mention the dry bias correction when describing the RAOB data, but it is unclear if you applied any correction, and if so which one?

We used the radiosondes sounding provided by the ARM archives (www.arm.gov). The radiosondes are RS92 and are processed using the standard Vaisala processing. No additional correction was applied to the radiosondes. We added text (Subsection 2.1) to better explain this point.

You present a short data segment from March 26, 2014, in Figure 2. Why did you select this specific period? Is it a typical period, or perhaps a period that is very stable (with low variability)?

Yes, the selected segment was a clear-sky night time segment with low vapour variability. We added more explanation in section 3 to explain this better.

You say that the different height of the GPS site was accounted for (page 7), but you do not explain how

This sentence, as well as a similar sentence on page 4 were actually left over from a previous version of the manuscript where we had used radiosondes from the station of Sodankylä which is at a higher elevation. In the current form the radiosondes from the station of Jyväskylä were used, to avoid any correction term. We thank the reviewer for this comment that prompted us to review the site locations and verify our calculations. The elevation of the instrumental 3 sites (Orivesi for GPS, Hyytiälä for MWRs & BAECC profiles, and Jyväskylä for the long-term RAOBs) were double checked using NASA/ASTER Digital Elevation Model data that has an estimated accuracy of ~ 15 m, and were found to agree in within 10 meters. Therefore, no corrections were applied. The sentences were removed.

**Technical Corrections**
page 1, line 13: In this work, we examine –> We examine -Done

page 1, line 26 (and many other places): there shall be a space between the value and the unit (also the % unit) according to SI rules. –Done, thank you

page 3: you may want to mention that the 89 GHz frequency is not only more sensitive to water vapour, but also to liquid water (clouds) which is a potential problem. This text could perhaps make more sense in Subsection 2.1 where the MWR is described.

Thank you we added a mention of this. The liquid water is accounted for when the retrievals are derived by simulating clouds in the column.

page 3, end of introduction: it is common practise to refer to the section numbers when the structure of the papers is presented. It is helpful for the reader.

We modified the last paragraph of Section 1 to provide a short overview of the sections of the paper.

page 3: A better title for Subsection 2.1 could be "Microwave radiometers and radiosondes" ? It will match the title of Subsection 2.2. -Done

page 4, line 30: the lunch times seem strange? I expect 6 h between the launches?

The 00 hours is actually 23. The time of the sonde launch generally varies between 15 and 35 minutes after the hour. For example, on 04/09/2014 the actual sonde launch was at 5:30, 11:16, 17:19, 23:20, however the exact time of the launch varies daily. We changed 00 to 23.

page 5, line 17: in zenith delay –> in the zenith delay - Done

page 7, line 29: low differences –> low mean differences? smallest differences –> smallest mean differences? -Changed

page 8, line 7: Differences –> Mean differences  - Changed

page 8, line 10: representativeness errors due to –> the true differences caused by

We changed this to "differences in the observed air masses caused by"

page 9, line 22: one RAOB –> RAOBs? -Done

Figure 1: I suggest to remove the text from the picture frame and describe which instruments (from left to right) that are seen in the figure caption. You may also want to mention the third instrument, the one to the right?

Done, we have removed text and added information in the caption

Figure 3: It is very difficult to see more that one of the time series. An alternative would be to show only one time series for the IWV and present the others as as differences from this one in individual subgraphs below.

This is a very good suggestion. We followed the reviewer's suggestion and left the GPS time series in the top panel. In the bottom panel we show the differences between GPS and the other instruments. This shows the increased variability during the months of June and July probably due to increased cloudiness and variability of the water vapor. In the new version of the manuscript, Figure 3 ->Figure 4.

Figure 4: I would prefer to have the important quantitative information either in the figure caption or in a separate table, if you regard it as important.

Done. Following both reviewers' suggestions added Table 3 that contains the quantitative information. In the new version of the manuscript, the Figure4 -> Figure 5.

Figure 7: Some of the red squares are hard to see. Plot them with larger symbols and perhaps a more light red colour. And since they are few, plot them on top of the black circles.

Done, they were already on top, but because of the color they were not visible. In the new version of the manuscript, Figure 7 -> Figure 8.

Figure 8: Mean and standard deviation of the measurement of each instrument –> RMS differences for each instrument pair vs the mean IWV.

The right side of Fig. 8 was not correct. We re-plotted it. It represents the standard deviation of the water vapor in the FOV of each instrument in each water vapor bin. In the new version of the manuscript, Figure 8 -> Figure 9.

---

## Author Comment (AC2) · 5 Nov 2018

**Response to interactive comment of Referee #2 on "Experimental total uncertainty of the derived GNSS-integrated water vapour using four co-located techniques in Finland" by E. Fionda et al.**

**Anonymous Referee #2**
**In black => referee observations**
**In red  =>  our response**

General comments

This study compares IWV estimates measured with 4 different instruments in Finland over a period of 9 months. Three of the instruments (2 microwave radiometers and a radiosonde system) are collocated and the fourth one (GPS) is distant by 20 km. The paper presents a statistical analysis of IWV differences and discusses the results in terms of inter-system biases and random errors including "representativeness errors". The study reports a dry bias in the radiosonde data during day, a bias source in the 2-channel radiometer data, and a tendency in the GPS IWV differences to increase when IWV increases. The latter effect is explained as "representativeness errors" due to the 20-km distance between the GPS receiver and the other systems. Emphasis is made on this result but two other factors are expected to contribute as well to the observed results: differences in the sensitivity and volume of sensed atmosphere between GPS and the other techniques and instrumental errors specific to each technique. Since these factors are not quantified, it seems difficult to assess the third one. A rigorous way of evaluating the impact of the distance between remote techniques requires using identical instruments at both ends. The discussion and conclusions of this study are impaired by this deficiency. To overcome this major weakness, the authors could consider the following options:

(a) The differences in IWV due to a 20-km distance can be evaluated from two identical instruments (e.g. a pair of GPS receivers) separated by approximately this distance in the same climatic region;

(b) The differences in IWV due to the different sensitivity and volume of sensed atmosphere between GPS and other techniques can be evaluated from co-located measurements (e.g. from past campaigns).

We thank the reviewer for this thoughtful comment. We agree that the effect of different volumes sampled by the instruments are likely to be one major source of uncertainty and were not properly discussed in the work. For this reason, we have substantially reviewed the analysis and added a new subsection (3.1) with a discussion of the different volumes of atmosphere sampled by the three techniques. Table 2 and Fig. 3 were also added in which the different field of views are specified

With regard to the two options proposed by the reviewer, option (a) is difficult to implement because GPS networks are sparse, with distances of the order of a hundred kilometers. To try to address the second option (b) we added in Table 3 previous findings from studies where GPS, radiometers, and radiosondes were collocated. Unfortunately, a GPS receiver is not part of the AMF2 and therefore it was not co-located during this campaign.

We think the additional analysis and discussion make the paper stronger and we thank the reviewer for the suggestions.

Specific comments

Give a clear definition of the "representativeness errors" in the introduction (Page 2 Line 27). Strictly

speaking, it should include the differences in the sensitivity and volume of sensed atmosphere between GPS and other techniques and the instrumental errors specific to each technique (see e.g. Buehler, AMT, 2012).

We agree with the reviewer. In the revised version we specifically refer to variability in the field of view and variability due to the fact that the instruments are not all strictly co-located.

The MWR retrieval algorithm used radiosonde data from a station located 90-km apart. Could there be a "representativeness error" impacting the results? Did you compare the operational radiosonde observations to those from the BAECC campaign used in this study?

Retrieval coefficients are usually derived for convenience from a radiosonde ensemble in a representative climatic region. However, they could as well be derived from a set of simulations (for example temperature and humidity profiles from a regional model). As long as the broad climatic region is representative of the radiometer's location the retrieval coefficients should be well defined. Because we used more than 10 years of radiosondes data, the regional atmospheric variability should be well captured. In the figures below the vapor density and temperature profiles of the training dataset (Jyväskylä, ~8000 profiles, blue) are plotted together with the BAECC profiles (~800 profiles, red). The climatology of the training Jyväskylä dataset is well suited to represent the AMF2 location. Specifically, in the year 2014 the mean IWV in Jyväskylä was 13.1 kg/m2 vs. 13.6 kg/m2 in Hyytiälä.

[Figure]

Can the linearity limitations observed with the 2-channel radiometer be explained/corrected? A temperature dependence is mentioned in the conclusions section though it is not discussed earlier. Please discuss this point in the text.

We added a discussion to this point in the text. It is our opinion that the slight temperature dependence observed in the 2CH data is due to the fact that the receiver of WVR1100 models are stabilized to 0.2 K (compared to the 30 mK of the 3CH). The antenna subsystem including the feedhorn is not thermally stabilized. Although a correction for residual temperature dependence of the parts that are not thermally stabilized (lens, feedhorn and waveguide) is routinely applied, there may be small residual dependencies.

Give more details about the BAECC radiosonde data. What is the vertical resolution of the profiles? How are they quality-controlled? Is any data processing/filtering or bias correction applied?

The RS92 radiosonde data used for the BAECC campaign were downloaded from the ARM Archive. It is our understanding that the data are processed with the standard Vaisala-provided algorithm and no additional corrections are applied. The vertical resolution is about 10 m and usually data are transmitted up to an altitude of 20 km. Standard quality control flag are applied and data quality

reports are issued to flag data that are considered incorrect. We added 2 references (Peppler at al, 2008, 2016) that explain in detail the quality control process at the Archive. We added these details in the text at the end of section 2.1.

Since the radiosonde measurements were made with a RS92 sonde type, did you consider applying the GRUAN bias corrections?

Although we are aware of the GRUAN program we are not familiar with the details of the GRUAN correction, therefore no ad-hoc corrections were attempted. It would be interesting to see, though.

How are the IWV data adjusted for the differences in altitude? (P5 L21)

This sentence was left over from a previous analysis were radiosondes from the station of Sodankylä were used to derive the coefficients. Sodankylä being at 180 m elevation, the radiosondes were extrapolated to the radiometer's elevation before deriving the coefficients. However, the Jyväskylä radiosonde station (which is the nearest available) is approximately at the same elevation as the GPS and radiometers stations at Orivesi and Hyytiala respectively. The elevations of the 3 sites were double checked using data from the NASA/ASTER Global Digital Elevation Map that has a quoted accuracy of 15 m and were found to agree with each other in within 10 meters. Therefore, no corrections were applied.

Why is the GPS conversion factor (pi) used as a monthly mean value and not from a linear regression with 2-m temperature following the method of Bevis et al. 1992? The uncertainty in the GPS IWV estimates could probably be reduced applying the Bevis method or time-varying Tm estimates from a NWP model.

The methodology applied in this work may actually reduce the uncertainty in the GPS IWV estimate when accurate temperature measurements are not available at the GPS location. Bevis' uncertainty estimate of about 2% was mostly attributed to uncertainty in the calculation of the weighted mean temperature of the water vapour which is dependent on surface temperature.

P. Basili et al (2001) proposed an alternative statistical approach, in which monthly average values of π were calculated from a historical database of radio-soundings. The estimated relative uncertainty for this approach was also in the 1.6 - 2 % range.

In absence of the surface temperature data at the GPS station and, because we are not sure of the uncertainty of surface temperature in NWP models the statistical approach is preferred in this specific case. Supplemental material is provided at the end of the responses to support this choice.

It seems to me that your GPS IWV data contain outliers. Figure 4 and Figure 7 show large deviations in the IWV differences which involve the GPS data. The outlying values should be removed from the analysis.

Although we agree with the reviewer that there are a few points that display higher scatter, we have no objective way to remove those points without introducing some arbitrary criteria. Following the suggestion of the first reviewer Fig. 3 (now Fig. 4) was modified to show the time series of the differences between the water vapor measurements. The largest discrepancies are during June and July when we expect conditions of increased cloudiness and variability. Rather than removing the points we have added a brief discussion in section 3.2.

The central description and discussion of results in section 3 should be supported by a statistical analysis including specifically the contributions from atmospheric variability (temporal/spatial) and

the different sources of differences (representativeness, etc.) between measurements (see e.g. Ning et al., AMT, Vol. 9, p79-92, 2016).

To address the comment, the following changes were implemented: 1) Added section 3.1 where the different volumes sampled by the systems are discussed. Added Table 2 where the FOVs (fields of view), the estimated total uncertainty of each instrument from error propagation, and the variability of each instrument while looking at a night scene with little water vapor variability are reported.

Fig. 3 was added which shows pictorially the volumes sampled by the systems. Although it is difficult to separate the contribution to the differences due to the FOVs and to the distance between instruments we provided some support to our conclusions that the distance of 20 km is contributing to the increased variability by adding in Table 3 findings of previous studies where similar instruments were co-located. In the 3 studies reported (Basili, Liljgren, Mattioli) the standard deviation of the differences between the GPS and the radiosondes (or radiometers) was similar to the one between the sondes and the radiometers which supports the claim that the much larger variability between the GPS and the other instruments found in this study is due to the physical distance between the sensors.

According to your interpretation of the results, you assume (implicitly) that the differences are random and independent (e.g. between GPS and MWR3C). Can you bring evidence of this?

Proving randomness and independence is not easy, we base our assumption on the fact that the instruments are independent and measure water vapor based on very different physical principles, however they all are measuring a quantity that is highly correlated in space and time.

It is not clear how you estimate the representativeness errors contribution (P8 L12-14; P9 L17-19) Given the above assumptions, the standard deviations should be added/subtracted quadratically.

We have revised the definition of representativeness error in the manuscript and modified the discussion. Because it is not possible to entirely separate the single contributions to the uncertainty, we have rephrased the conclusion to refer to a total uncertainty that includes a contribution from the FOV and a contribution due to the distance from the GPS.

Figure 8 left panel: it might be more interesting to plot the SD rather than the RMSD to support the above discussion.

Done and as a consequence the definition of RMSD (eq 7) was removed as this quantity was not used anymore. In the revised manuscript Figure 8 -> Figure 9

In the conclusion section, it is stated that the comparison of the two radiometers gives confidence in the "stability" of these measurements but this point is not discussed earlier in the paper. Can you elaborate a bit more on this?

We have not observed any drifts or dependency of the mean differences on time. For this reason, we think all the instruments performed nominally throughout the campaign. Following the comment from Reviewer #1 this sentence was deleted from the revised version.

Technical comments

P1 L25 and 26: you refer incorrectly to IWV "variability" above or below 20 kg/m2 in place of simply "IWV". Please correct. -Done

P6 The quantity defined by equation (5) should be referred to as the "mean difference"  -Done

P7 L23-24: the comment about the log-normal PDF is not used later, this sentence can be removed. -Done

P7 L25: the comment on the agreement between measurements is elusive. Please remove it. -Done

P8 L9: the GPS and MWR3C are "unbiased" maybe you want to write "have a mean different close to zero". To prove that an instrument is unbiased requires to compare its observations to a calibrated reference standard. However, you may argue here that the instruments provide independent observations which may have specific biases, however it is unlikely that their biases would perfectly cancel out, so it is much more likely here that they are both unbiased.

We agree with the reviewer comment. What we really meant is that they have no bias relative to each other as we can't prove that they are truly unbiased.

P8 L15: "higher variability in summer": this is not obvious from Figure 9 right panel.

We thank the reviewer for this comment. Fig. 8 was not showing the correct data. We replotted the figure (now Fig. 9) that now shows the higher variability of the data with increasing water vapor. From Fig. 4 it can be seen that the higher variability occurs in June and July.

P8 L16-19: you suggest to include a "representativeness" error of 1-2 kg/m2 to account for the spatial variability of GPS measurements. How would you do that? Shouldn't this error depend on the distance to the GPS measurements?

Yes, we have rephrased. In this case it seems that because of the distance between the instruments the different volumes sampled by the systems resulted in larger variability of the differences.

P8 L20-23: the comment of the bottom panel of Fig. 6 doesn't add any further information about the radiosonde bias at 11 UTC. Remove this comment as well as the bottom panel of Fig. 6 which shows merely a combination of the lines plotted in the upper panel.

If the reviewer has no objection we rather remove the top panel and keep the bottom panel of Fig 6 that shows the bias compared to all systems. In the revised manuscript, Figure 6 -> Figure 7

P9 L21-31: these two paragraphs are not useful in the conclusion section.

-Removed

Figure 3: the series are superposed and cannot be distinguished. Maybe show only one series to illustrate the IWV variations over the study period.

-Done. In the revised manuscript Figure 3 -> Figure 4

Figure 4: it is common practice to draw the common reference data set (here GPS) on the x-axis. Please change the 3 plots.

-Done

It may be useful to add a scatter plots of RAOB vs. MWR3C and MWR2C vs. MWR3C. -Done

**Supplemental information on the conversion of ZWD to IWV using a statistical conversion factor π**

It is possible to retrieve the IWV from ZWD calculated by GNSS observation using the well-known following relationship:

$$IWV = \pi * ZWD \tag{1}$$

The conversion coefficient π is a "constant" of proportionality which depends on several physical parameters, among which the weighted mean temperature of the water vapor (Tm):

$$Tm = \frac{\int (e/T)dz}{\int (e/T^2)dz} \tag{2}$$

$$\pi = 10^6 / \rho_l R_v \left[ \left( k_3 / T_m \right) + (k_2 - mk_1) \right] \tag{3}$$

where $\rho_l$ is the density of liquid water, $R_v$ is the specific gas constant for water vapor, $m$ is the ratio of molar masses of water vapor and dry air, $k_1$, $k_2$, $k_3$ are constants (77.6036, 64.79, $3.77 \times 10^5$), $T$ is the air temperature, $e$ is the partial pressure of water vapor, and dz has units of length in the zenith.

**Computation of the conversion coefficient**
Eq. 1 assumes that the Wet Delay is entirely due to water vapour and liquid water traveled by the GNSS signal to reach the ground receiver [1]. Bevis et al 1994 [2] showed that in most practical conditions the uncertainty of π is mainly due to Tm. When Tm is predicted from surface temperature on the basis of a linear regression, Bevis estimated a relative error in π of the order of 2%.

P. Basili et al (2001) [3] proposed an alternative statistical approach, in which monthly averaged π values were calculated from a "historical" database of radio-soundings. This approach was developed to overcome the lack of surface temperature measurements at many GPS sites. The approach was tested using a GPS from Cagliari, Italy. The first two columns of Table S1 report the computed values of π with associated uncertainties computed for each month. It can be noticed that the relative error (Std. Dev., %) of the monthly averages ranges between 1.14 % (October) and 1.61% (January), and using a constant value of π = 0.152 (yearly value) produces a relative error of 2.15 %.
In the same work π was computed following the Bevis approach by computing Tm with a linear regression on surface temperature. The relative error of π in this case was equal to 1.33%, which is in the same range (1.14 % - 1.61 %) of the errors obtained by the statistical approach (see Table S1).

Due to the lack of surface temperature data at the GPS station, the statistical approach proposed in [3] was adopted in this work. The performance of the chosen approach was analyzed looking at two long-term RAOB Finnish databases in Sodankylä (2000-2014) and Jyväskylä (2003-2014). Supplemental Table S1 reports the computed statistical monthly conversion coefficient (π) values with associated uncertainties at the two stations. The results are similar to those for Cagliari with average monthly relative error ranging between 1.24 % and 1.76 % in Sodankylä (2.72 % when using all year values) and between 1.15% and 1.83 % in Jyväskylä (2.77 % when using all year values). Supplemental Figure S1 shows the seasonal dependence of the conversion coefficient at the 3 sites.

Based on these results a statistical approach can provide good estimates π when climatologic information is available and surface temperature measurements are not available at the GNSS station.

Table S1 – Statistical conversion coefficient (π) values for Cagliari [3], Sodankylä, and Jyväskylä

| Months | Cagliari (I) | | Sodankyla (FIN) | | Jyvaskyla (FIN) | |
| --- | --- | --- | --- | --- | --- | --- |
| | π | Std. Dev. (%) | π | Std. Dev. (%) | π | Std. Dev. (%) |
| 1 | 0.149 | **1.61** | 0.147 | **1.62** | 0.148 | **1.74** |
| 2 | 0.148 | **1.48** | 0.147 | **1.69** | 0.148 | **1.83** |
| 3 | 0.149 | **1.39** | 0.147 | **1.60** | 0.149 | **1.80** |
| 4 | 0.150 | **1.23** | 0.150 | **1.52** | 0.151 | **1.44** |
| 5 | 0.152 | **1.20** | 0.152 | **1.76** | 0.154 | **1.68** |
| 6 | 0.154 | **1.29** | 0.154 | **1.53** | 0.156 | **1.44** |
| 7 | 0.156 | **1.17** | 0.157 | **1.24** | 0.158 | **1.15** |
| 8 | 0.156 | **1.24** | 0.156 | **1.27** | 0.157 | **1.26** |
| 9 | 0.154 | **1.22** | 0.154 | **1.38** | 0.155 | **1.26** |
| 10 | 0.153 | **1.14** | 0.151 | **1.62** | 0.153 | **1.54** |
| 11 | 0.151 | **1.51** | 0.149 | **1.54** | 0.151 | **1.62** |
| 12 | 0.149 | **1.42** | 0.148 | **1.66** | 0.149 | **1.67** |
| **All years** | **0.152** | **2.15** | **0.151** | **2.72** | **0.152** | **2.77** |

[Figure]

Fig. S1 – Monthly mean of the statistical conversion coefficient (π), calculated at black: Sodankylä, red: Jyväskylä, and blue: Cagliari.

References
[1] J. Duan, M. Bevis, P. Fang, Y. Bock, S. Chiswell, S. Businger, C. Rocken, F. Solheim, T. Van Hove, R. Ware, S. Mclusky, T.A. Herring, and R.W. King, "GPS meteorology: Direct estimation of the absolute value of precipitable water", J. Appl. Meteorol. vol. 35, pp. 830-838, 1996.
[2] M. Bevis, S. Businger, S. Chiswell, T. A. Herring, R. A. Anthes, C. Rocken, and R. H. Ware, "GPS meteorology: Mapping zenith wet delays onto precipitable water," J. Appl. Meteorol., vol. 33, pp. 379–386, 1994.

[ 3]   P. Basili, S. Bonafoni, R. Ferrara; P. Ciotti, E. Fionda and R. Ambrosini, "Atmospheric water vapor retrieval by means of both a GPS network and a Microwave Radiometer during an Experimental campaign in Cagliari, Italy, in 1999", IEEE Geoscience Remote Sensing, vol. 39, N.11, pp. 2436-2443, Nov 2011.

[4]   M. Bevis, S. Businger, T. A. Herring, C. Rocken, R. A. Anthes, and R. H. Ware, "GPS meteorology: Remote sensing of atmospheric water vapor using the Global Positioning System," J. Geophys. Res., vol. 97, pp. 787–801, 1992.

[5]   O. T. Davies and P. A. Watson, "Comparison of integrated precipitable water vapor obtained by GPS and radiosondes," Electron. Lett., vol. 34, pp. 645–646, 1998.